# Microbiota Transplant and Gynecological Disorders: The Bridge between Present and Future Treatments

**DOI:** 10.3390/microorganisms11102407

**Published:** 2023-09-27

**Authors:** Serena Martinelli, Giulia Nannini, Fabio Cianchi, Fabio Staderini, Francesco Coratti, Amedeo Amedei

**Affiliations:** 1Department of Clinical and Experimental Medicine, University of Florence, 50139 Florence, Italy; serena.martinelli@unifi.it (S.M.); giulia.nannini@unifi.it (G.N.); fabio.cianchi@unifi.it (F.C.); fabio.staderini@unifi.it (F.S.); corattif@gmail.com (F.C.); 2SOD of Interdisciplinary Internal Medicine, Azienda Ospedaliera Universitaria Careggi (AOUC), 50139 Florence, Italy

**Keywords:** microbiota transplantation, vaginal microbiota, gynecological disorders

## Abstract

Fecal microbiota transplantation (FMT) is a procedure that involves transferring fecal bacteria from a healthy donor to a patients’ intestines to restore gut–immunity homeostasis. While FMT was primarily supposed to treat gastrointestinal disorders such as inflammatory bowel disease and irritable bowel syndrome—and especially Clostridium difficile infection (currently the only used as clinical treatment)—recent research has suggested that it may also become a potential treatment for gynecological disorders, including endometriosis and polycystic ovary syndrome (PCOS). On the contrary, vaginal microbiota transplantation (VMT) is a newer and less commonly used procedure than the FMT approach, and its potential applications are still being explored. It involves direct grafting of the entire vaginal microbiota of healthy women into the vaginal tract of patients to easily rebuild the local microbiota environment, restoring vaginal eubiosis and relieving symptoms. Like FMT, VMT is thought to have potential in treating different microbiota-related conditions. In fact, many gynecological disorders, such as bacterial vaginosis and vulvovaginal candidiasis, are thought to be caused by an imbalance in the vaginal microbiota. In this review, we will summarize the development, current challenges, and future perspectives of microbiota transplant, with the aim of exploring new strategies for its employment as a promising avenue for treating a broad range of gynecological diseases.

## 1. Introduction

Humans are ‘superorganisms’ that are colonized by 10^14^ total cells of more than 1000 species of bacteria, especially in the gastrointestinal tract [1,2]. Each host has a unique microbiota, which is essential for its immunity, nutrition, and pathogenesis [3,4]. Microorganisms inhabit various anatomical sites in the body such as the skin, mucosa, gastrointestinal tract, respiratory tract, urogenital tract, and mammary gland. They establish complex and distinct ecosystems that adapt to the specific environmental circumstances of each niche [5].

Starting from birth, a strict symbiotic relationship between the human body and its native microbiota is established. This relationship plays crucial functions in maintaining overall health and well-being. Therefore, its pivotal roles in protecting against pathogens; regulating metabolic, endocrine, and immune processes; and in influencing drug metabolism and absorption have started to be elucidated [5]. The microbiota leads the body to experience various changes from conception to death and also undergoes continuous modifications throughout life, which are influenced by various host elements such as age, dietary habits, lifestyle, hormonal fluctuations, and medical conditions [6]. Nevertheless, alterations in the microbiota composition, known as dysbiosis, have the potential to result in severe and even life-threatening illnesses, including gynecological disorders [7,8]. In detail, the gut microbiota (GM) functions as an endocrine organ that can influence other distant organs such as the central nervous system and the liver. It has been demonstrated that many chronic conditions, such as obesity, diabetes, and non-alcoholic fatty liver disease (NAFLD), are all linked to GM dysbiosis [9]. Recent investigations suggest that intestinal dysbiosis can affect gut permeability, the innate immune system, the fermentation of indigestible carbohydrates, and the intestinal production of short-chain fatty acids (SCFAs), which can lead to NAFLD [10,11]. Consequently, NAFLD is related to other diseases like diabetes mellitus, obesity, metabolic syndrome, hypertension, renal disorders, and cardiovascular diseases—pathologies that strongly affect human life [12,13,14]. 

SCFAs are a group of fatty acids produced in the colon by the bacterial fermentation of dietary fibers and resistant starch. The most common types are butyrate, propionate, and acetate. SCFAs allow the growth of homeostasis-promoting bacteria, such as *Lactobacilli* and *Bifidobacteria*, and hinders colonization by opportunistic pathogenic bacteria, including *Clostridium* and *Escherichia coli* [15]. In addition, SCFAs contribute to the preservation of a functional gut barrier and to the maintenance of host homeostasis by stimulating the regeneration of epithelial cells and the production of mucus and antimicrobial peptides [16]. Other critical SCFAs roles are the modulation of T regulatory (Treg) cells, as well as their crucial physiological effects on several organs, including the brain and the urogenital apparatus [17,18,19]. G-protein coupled receptors (GPCR) and free fatty acid receptors (FFA) are the main receptor types that are activated by SCFAs; they are expressed in several tissues and regulate both energy metabolism and immune response [20]. Growing evidence suggests that intestinal dysbiosis can cause various immune and metabolic alterations through the activity of bacterial products: a decrease in SCFAs’ production can increase the risk of intestinal, neurological, and cardiovascular diseases development [21,22,23,24]. Dysbiosis can also increase lipopolysaccharide (LPS) circulating levels that can enhance the inflammatory response through micro-organism-associated molecular pattern (MAMPs) and the activation of pattern recognition receptors (PRR) such as toll-like receptors (TLRs) signaling pathways [25,26] (Figure 1). Moreover, the production of pro-inflammatory cytokines, including interleukin-17 (IL-17), tumor necrosis factor-α (TNF-α), and interferon-γ (INF-γ), can result in reactive oxygen species (ROS) release in the cells of distant organs, including the gynecological tract, whose homeostasis could be altered, contributing to the induction of a pathological state [27] (Figure 1). In addition to SCFAs, there are several metabolites produced by the microbiota that are involved in the gut-brain axis modulation, such as neurotransmitters [28] (Figure 1).

This work aims to review and discuss the impact of fecal and vaginal microbiota transplantation in manipulating microorganism functions, leading to the improvement of human gynecological disease and health maintenance.

## 2. Methods

We conducted a PubMed search for original articles, reviews, meta-analyses, and case series using the following keywords, their acronyms, and their associations: gut microbiota, vaginal microbiota, fecal microbiota transplantation, vaginal microbiota transplantation, short chain fatty acids, gynecological disorders, polycystic ovary syndrome, endometriosis, bacterial vaginosis, microbiota-changing strategies, and nutrition. The items found from the above-mentioned sources were reviewed by two of the authors (S.M. and G.N.), and a narrative review was performed.

## 3. Role of the Gut Microbiome–Estrogen Axis in Gynecological Disorders

It has been widely recognized that the mutual relationships and metabolic activities among microorganisms have a significant impact on the host pathophysiology [28]. In particular, the gut microbiome (GM) has been found to modulate hormone levels in the body, especially estrogens in women [29]. The connection between GM and estrogen was first observed thirty years ago when Adlercreutz et al. found that antibiotics assumption reduced estrogen levels in women [30]. The GM mainly controls estrogen levels through the secretion of an enzyme called β-glucuronidase, which is encoded by several GM genera, including *Bacteroides*, *Bifidobacterium*, *Escherichia*, and *Lactobacillus* [31]. This enzyme converts conjugated estrogens to deconjugated forms in the gastrointestinal tract. These deconjugated and unbound “active” estrogens enter the bloodstream and subsequently act on estrogen receptor alpha (ERα) and estrogen receptor beta (ERβ), eliciting downstream activation of intracellular signaling cascades, gene transcription, and epigenetic effects [29,32]. When there is a decrease in β-glucuronidase activity due to an imbalance in the GM community (dysbiosis), there is less estrogen deconjugation, resulting in lower circulating estrogen levels [33]. Conversely, increased β-glucuronidase activity can increase estrogen levels. Thus, maintaining optimal β-glucuronidase activity is critical for regulating estrogen levels in females [34]. 

Estrogens contribute to epithelial proliferation throughout the female reproductive system and have been shown to drive proliferative diseases such as endometriosis and polycystic ovary syndrome (PCOS) [8,35]. 

Endometriosis is defined as the presence of endometrial tissue outside the uterus, including glands and stroma, that express the ER and therefore respond to estrogen. Endometriosis is a benign chronic inflammatory gynecological disease, but it can also involve the malignant behavior of invasion and migration [36]. Some studies have found higher levels of taxa that encode for β-glucuronidase, which are mainly *Bifidobacterium* and *Escherichia*, in endometriosis women compared with control women [37,38]. Wei Y. et al. found that β-glucuronidase promoted endometriosis development directly or indirectly by causing macrophage dysfunction. They documented the GM changes on patients and mice with endometriosis and the effect of β-glucuronidase on the proliferation and invasion of endometrial stromal cells and the development of endometriotic lesions [39]. 

The correlation between GM and hormones regulation was also found to affect PCOS disease [40]. PCOS is a common endocrine disorder in women of reproductive age and its clinical features are mainly oligo-ovulation or anovulation, hyperandrogenemia and insulin resistance. Moreover, PCOS is considered one of the leading causes of infertility in women of childbearing age [41,42]. The etiology and pathogenesis of PCOS remain unclear and may be multi-factorial, but in recent years growing evidence has highlighted a GM role in modulating PCOS progression. 

In 2012, Tremellen et al. hypothesized that the gut flora could be related to PCOS, suggesting that GM imbalance could be associated with various PCOS clinical symptoms, such as hyperandrogenemia, multiple ovarian cysts, and anovulation [43]. Since then, several studies have found that specific microflora was changed in PCOS patients compared to healthy controls, such as the variation in the balance between *Bacteroides* and *Firmicutes* [40,44,45]. Moreover, the increase in the relative abundance of *Firmicutes* and *Bacteroidetes* was positively correlated with androgens concentration, body mass index, and insulin resistance, as well as with the level of free testosterone [46,47]. Regarding the genera *Bacteroides*, Liu et al. observed an increase in *Escherichia* and *Shigella* in PCOS patients, and a similar GM composition compared to obese control women [48]. The GM modifications in PCOS are different, sometimes controversial, and not yet fully understood.

Anyway the GM may influence the sex hormones levels also by the production of SCFAs [49]. These SCFAs have been shown to exert anti-proliferative effects [50,51] and have some anti-inflammatory properties that can be extended to distant organs [52]. SCFAs primarily affect the cells through several mechanisms, including the activation of G-protein-coupled receptors, namely, GPR43, GPR41, and GPR109A [53], which are known to downregulate inflammation [54]; and the inhibition of histone deacetylases [55], and the inhibition of histone deacetylases [56]

In an interesting in vitro study, porcine granulosa cells treated with low concentrations of butyric acid showed increased progesterone secretion, while higher butyrate concentrations significantly inhibited the progesterone secretion via the cAMP signaling pathway [48]. Another study by Liu et al. showed that the gut-derived butyrate can contribute to nonalcoholic fatty liver disease in premenopausal women with estrogen deficiency [56]. Moreover, it has been observed that fecal samples of mice affected by endometriosis contained lower levels of SCFAs and butyrate compared with control mice, and treatment with butyrate resulted in a decrease in the growth of both mouse and human endometriotic lesions [57,58].

There is evidence of lower concentrations of SCFAs in fecal samples from PCOS patients [59]. Indeed, probiotics’ supplementation promoted the growth of *Faecalibacterium prausnitzii*, *Bifidobacterium*, and *Akkermansia*, which are SCFA-producing bacteria, and can lead to an increase in intestinal SCFAs. In turns, SCFAs bind to their receptors on enteroendocrine cells and directly stimulate the release of gut–brain mediators that can influence sex hormone secretion by the pituitary gland and hypothalamus via the gut–brain axis [60] (Figure 2).

These studies confirm that the GM metabolites produced in response to dietary intake may play a role in regulating estrogen and progesterone levels in women. In light of these data, GM relevance in female gynecological disorders and in conditions affecting the reproductive tract must be taken into consideration [61,62]. 

While investigations in mice are starting to propose ways in which the GM impacts female gynecological disorders, no causal relationships between the GM and such disorders have been established in humans [33]. The significance of specific GM metabolites derived from the gut in relation to female reproductive health remains unknown. However, this field of study holds promise as it could lead to dietary interventions and/or fecal microbiota transplantation to improve the impact of gynecological disorders.

## 4. Fecal Microbiota Transplantation

Over time, there has been a growing fascination regarding unraveling GM composition and devising approaches to manipulate it. The initial phase of comprehending the effects on the equilibrium between health and disease lies in characterizing the composition of intestinal microbes, paving the way for the development of tailored interventions for specific clinical conditions. 

Fecal microbiota transplantation (FMT) is the clinical approach of transferring human fecal microbiota (the entire microbial community) from a healthy donor to a recipient gastrointestinal tract to treat a disease related to microbiota imbalance and to restore the overall microbial diversity and host intestinal healthy [63]. 

The history of FMT dates back several centuries, although its scientific understanding and clinical application have evolved over time: FMT-like practices were reportedly being used in ancient China in the fourth century. A text titled “Medicine’s Golden Mirror” described the use of “yellow soup” prepared with fecal matter as a treatment for severe diarrhea [64]. During the 17th century, some European physicians tried similar approaches. A veterinarian named Christian Franz Paullini has documented the use of oral administration of fecal matter from healthy animals to treat dysentery in horses [65]. 

In the 20th century, the first modern reference to FMT was made in 1958 when a physician named Ben Eiseman successfully treated four patients with fulminant pseudomembranous enterocolitis (now known as Clostridium difficile infection) using fecal enemas [66,67]. In the 1980s, FMT gained attention as a potential treatment for recurrent CDI (Clostridium difficile infection). Dr. Thomas Borody, an Australian gastroenterologist, published case reports demonstrating the efficacy of FMT in treating patients with severe and recurring diarrhea [68]. Further research on the use of FMT for CDI in the early 2000s reported high success rates in treating recurrent CDI, sparking further interest in FMT as a treatment option [69]. In 2013, the first randomized controlled trial (RCT) evaluating FMT for recurrent CDI was published. The study confirmed FMT’s efficacy, with a success rate of 94% compared to 31% in the control group receiving antibiotics alone [70]. In 2013–2019, FMT gained recognition and acceptance in clinical guidelines. The U.S. Food and Drug Administration (FDA) released guidelines stating that FMT could be considered for recurrent CDI that does not respond to standard therapies [71]. Since 2020, the has research expanded to explore the potential of FMT in inflammatory bowel disease (NCT04436874), irritable bowel syndrome, metabolic disorders, and in patients affected by various other diseases, such as acute graft-versus-host disease (NCT03492502 Clinicaltrials.gov), in patients with malignancies not responding to immune checkpoint inhibitor therapy (NCT05273255), and in patients affected by amyotrophic lateral sclerosis [72]. Indeed, there are more than 400 ongoing clinical trials to evaluate the efficacy of FMT in various pathologies (complete list available on Clinicaltrials.gov) including female urogenital diseases with (NCT03786900) or without (NCT03786900) pregnancy complications.

## 5. FMT and Gynecological Disorders in Animal Models

To date, there are no clinical reports of the use of FMT to treat gynecological disorders except in mouse models; prospective data from laboratory research should encourage further studies in humans [73]. 

In a PCOS rat model, *Lactobacillus*, *Ruminococcus*, and *Clostridium* fecal concentration were lower while *Prevotella* was higher than in control rats. The treatment of PCOS rats with *Lactobacillus* or FMT from healthy rats showed that estrus cycles were improved in all rats in the FMT group and in 6 of 8 rats in the *Lactobacillus* transplantation group, with a decreased biosynthesis of androgens, a normalization of ovarian morphologies, and restoration of intestinal microbiota composition in both groups, also showing an increase in *Lactobacillus* and *Clostridium* and a decrease in *Prevotella*. This study thus demonstrated that GM dysbiosis was associated with PCOS pathogenesis, and that microbiota shaping through FMT or *Lactobacillus* administration were beneficial for the treatment of PCOS rats [74]. 

Torres PJ et al. performed a cohousing study using a letrozole-induced PCOS mouse model that recapitulates many of the reproductive and metabolic characteristics of PCOS. Because mice are coprophagic, cohousing results in repeated, noninvasive inoculation of gut microbes in cohoused mice via the fecal–oral route. In contrast to letrozole-treated mice housed together, letrozole mice cohoused with placebo mice showed a significant improvement in both reproductive and metabolic PCOS phenotypes [75]. Gene sequencing showed that the overall GM composition and the relative abundance of *Coprobacillus* and *Lactobacillus* were different in letrozole-treated mice cohoused with placebo mice compared with letrozole mice housed together, suggesting that GM dysbiosis may play a causal role in PCOS, and that its modulation may be a potential treatment option for it. 

Using a mouse model of endometriosis, Chadchan SB et al. demonstrated that treating the mice with antibiotics resulted in a decrease in endometriotic lesions. Furthermore, they observed a distinct difference in the GM composition between mice with endometriosis and those without. In detail, the mice with endometriosis showed a higher abundance of *Bacteroidetes* and a lower abundance of *Firmicutes*. Notably, when metronidazole (a drug targeting the *Bacteroides* genus) was administered, a decrease in the growth of endometriotic lesions was observed. However, interestingly, when the mice were orally given fecal matter from mice with endometriosis, the lesions’ growth was restored. These results show that gut bacteria play a significant role in promoting the progression of endometriotic lesions, and that shaping the intestinal bacteria by FMT can modulate disease progression [76].

Recently, successful therapies using FMT have inspired a new application of microbiota transplantation: vaginal microbiota transplantation (VMT), which is a medical procedure that involves the transfer of healthy bacteria from a donor to a recipient’s vagina [77]. This is a relatively new area of research and is being explored as a potential treatment for some vaginal conditions and imbalances, such as recurrent bacterial vaginosis [78].

## 6. The Human Vaginal Microbiota

As previously mentioned, many urogenital illnesses, including bacterial vaginosis and different infections (e.g., yeast, sexually transmitted, and urinary tract), appear to be prevented in part by the human vaginal microbiota [79,80,81]. From childhood to reproductive age and up until menopause, the lower reproductive tract (LRT) microbiota composition changes in females [82], and one of the key factors impacting these microbiota modifications is hormonal change [83]. 

In the pool of female reproductive tract microbiota, *Lactobacillus* spp. predominates in human females, while it accounts for less than 1% of the population in other mammals. The majority of vaginal communities (73%) were dominated by one or more species of *Lactobacillus*, unlike any other anatomical location of the human body. The LRT’s pH was exceptionally low (4.5) as a result of lactic acid, which is the main metabolic by-product of *Lactobacillus* in the presence of glycogen as substrate. 

In detail, the *Lactobacillus* spp. seems to play key protective roles by producing various bacteriostatic and bactericidal compounds, or through competitive exclusion [84,85,86]. There is a stronger correlation between reproductive eubiosis and certain microbial community, dominated by *Lactobacillus* spp. 

The cervicovaginal microbiota resides in and on the epithelium’s outermost layer. A healthy cervico-vagina exhibits the dominance of *Lactobacillus* spp. (10^7^–10^9^ *Lactobacilli*/gram of vaginal fluid), which can comprise up to 95% of the total bacterial population living in the entire reproductive tract. Several studies have been performed on the cervico-vaginal microbiota, and at least five categories, known as community state types (CSTs), have been identified [87,88]. Four CSTs categories exhibited dominancy of *Lactobacillus* spp. CST-I is dominated by *L. crispatus*, whereas CST-II, CST-III, and CST-V show dominancy of *L. gasseri*, *L. iners*, and *L. jensenii*, respectively. The fifth, CST-IV, has a lower density of *Lactobacillus* spp. [87]. 

The latter category is subcategorized in three subgroups; in particular, the first of these is defined by the cooccurrence of seven taxa, including *Cryptobacterium*, *Gemella*, *Gardnerella*, *Aerococcus*, *Prevotellaceae_1*, and *Ruminococcaceae_7*, and *Anaeroglobus*. The second cluster is also defined by the presence of seven taxa, including *Sneathia*, *Megasphaera*, *Anaerotruncus*, *Eggerthella*, *Atopobium*, *Prevotellaceae_2*, and *Parvimonas*. The third cluster is characterized by the presence of five taxa, including *Porphyromonas*, *Peptoniphilus*, *Mobiluncus*, *Dialister*, and *Prevotella*. Although they vary greatly in abundance, the majority of communities in groups I, II, III, and V contain multiple phylotypes of lactic acid bacteria, indicating some degree of functional redundancy. Species of *Lactobacillus* (groups I, II, III, and V) dominated the vaginal bacterial populations in 80.2% and 89.7% of Asian (pH 4.4 ± 0.59) and White (pH 4.2 ± 0.30) women, respectively, but only in 59.6% and 61.9% of Hispanic (pH 5.0 ± 0.74) and Black women (pH 4.7 ± 1.04). Comparing Hispanic and Black women to Asian and White women, the higher pH values in these two ethnic groups represent the higher prevalence of communities not dominated by *Lactobacillus* sp. (cluster IV) [87]. Although it is uncertain what causes these variations among ethnic groups, it is tempting to hypothesize that genetic differences between hosts may be a major factor in the species composition of vaginal communities. Variations in the innate and adaptive immune systems, the kind and volume of vaginal secretions, and ligands on the surfaces of epithelial cells could be affecting factors. However, research has demonstrated that human behaviors and habits, such as personal hygiene, birth control methods, and sexual behaviors, also have a relevant impact on the development of vaginal communities [89].

## 7. Vaginal Microbiota Transplantation and Gynecological Disorders

Gynecological disorders that have been explored in relation to VMT include bacterial vaginosis, vulvovaginal candidiasis (VVC), and specific cases of infertility. These conditions are often associated with an imbalance in the vaginal microbiota, characterized by a decrease in beneficial *Lactobacillus* species and an overgrowth of pathogenic bacteria or fungi. The use of microbiota transplantation from healthy women has been suggested as a potential therapy to address imbalances in the vaginal microbiota, known as vaginal dysbiosis. Studies in rat models of vaginal dysbiosis have demonstrated that VMT can be therapeutically effective in reducing inflammation and increasing the presence of Lactobacilli, as well as relieving endometritis-like symptoms (inflammation of the lining of the uterus) [90,91]. The benefits of therapeutic VMT have also been demonstrated in patients with symptomatic, intractable, and recurrent bacterial vaginosis, and this has opened a new avenue for such future studies [78].

Bacterial vaginosis (BV) refers to a condition in which there is a disturbance in the normal microbial community of the vagina. As well previously documented, usually, the vagina is predominantly inhabited by *Lactobacillus* species [92,93]. However, in BV presence, there is a shift in the vaginal microbiome, leading to the emergence of anaerobic bacteria [87,94,95]. BV may be associated with a risk of upper genital tract infections, pregnancy complications, and susceptibility to sexually transmitted infections [96]. Currently, there are many limited treatment options in patients with persistent or recurrent BV despite multiple attempts at antibiotic treatment [97]. In addition, probiotic treatment of symptomatic patients with oral and/or vaginal administration of bacterial strains of *Lactobacillus* has yielded conflicting results [98], suggesting that the microbiome as a whole, rather than a single bacterial species, may be effective for severe BV. 

In this scenario, Lev-Sagie et al. performed the first exploratory study testing the VMT approach from healthy donors as a therapeutic alternative for patients suffering from symptomatic, intractable, and recurrent bacterial vaginosis [78] (Clinicaltrials.gov NCT02236429). Four out of five VMT recipients experienced a significant improvement in both clinical symptoms and the composition and function of the dysbiotic vaginal microbiome, which persisted over an extended follow-up period, while one recipient experienced partial remission. The authors reported no significant side effects and no serious adverse events. While the lack of adverse outcomes is reassuring, the small study size and lack of a placebo arm make it difficult to interpret whether VMT provided an additional benefit over antibiotics alone. Additional clinical trials are currently ongoing with the goal of further evaluating whether VMT could serve as a viable option in symptomatic and intractable BV (Table 1). 

Vulvovaginal candidiasis (VVC) is a yeast infection caused by the overgrowth of the Candida species. Current studies on targeting vaginal dysbiosis by replacing with beneficial bacteria in women affected by VVC include oral administration of the specific probiotic strain *Lactobacillus plantarum* P17630. The treatment was associated with colonization of lactobacilli on the vaginal epithelial cells and with an improvement of clinical signs, such as flushing, swelling, and discharge [99]. In addition, long-term administration of lactobacilli-based probiotics, such as *Lactobacillus crispatus* and *Lactobacillus delbrueckii*, were found to inhibit 60–70% of *Candida albicans* in a Sprague Dawley rat model of VVC [100]. 

The effectiveness of oral or intravaginal administration of *Lactobacillus acidophilus*, *Lactobacillus rhamnosus* GR-1, and *Lactobacillus fermentum* RC-14 in colonizing the vagina and/or preventing vaginal colonization and infection by *Candida albicans* has been demonstrated in clinical trials [101]. Furthermore, lactobacilli have exhibited beneficial effects in the prevention and treatment of recurrent VVC when administered along with standard antifungal drugs [102]. 

Finally, in recent years, yeast-based probiotics have been proposed as a promising tool for the prophylaxis and treatment of vaginal infections, and *Saccharomyces cerevisiae* var. boulardii is the only probiotic yeast commercially available for human use [103,104]. Further, its therapeutic activity has also been demonstrated in clinical studies [105]. 

Before considering VMT as a clinical option for preventing recurrent gynecological disorders, it is essential to test the safety, feasibility, and effectiveness of the vaginal matter. For this reason, Yockey L et al. presented a protocol that outlines comprehensive safety screening procedures and the collection of vaginal fluid donations specifically for VMT. This protocol aims to establish the necessary groundwork to ensure the safety and viability of VMT as a potential intervention in clinical care [106].

## 8. Other Microbiota-Changing Strategies and Future Perspectives

In addition to FMT, the advantages of which are illustrated in Figure 3, there are other strategies such as diet, and the administration of prebiotics [107], probiotics [108], symbiotics (combination of probiotics and prebiotics) [109], and postbiotics [110,111]. Recent research has documented that more than 50% of the diversity in the microbiome can be attributed to food, while host genetics only slightly influence the composition of the microbiota [112]. In detail, a diet high in fiber promotes a significant rise in species producing SCFAs [113]. A high-protein, high-fat, low-fiber diet, on the other hand, is linked to decreased biodiversity and an increase in species that could cause inflammation [114]. A brief food change may cause a change in the gut’s population, but these modifications seem to be temporary. Diet may increase the success of FMT by fostering a favorable environment for the engraftment of donor microbiota and by exerting its own anti-inflammatory effects [115]. 

However, these natural approaches lack a direct-targeting action for GM shaping. For this purpose, researchers have developed the use of engineered bacterial (EB) strains to influence and manipulate the composition and behavior of the microbiota in a promising targeted manner [116]. EB strains can be designed to produce specific metabolites or molecules that, interacting with the microbiota, can influence its composition or activity, showing beneficial effects on human health. EBs are rightly considered the next-generation microbiota therapeutics [117,118]. In addition, they can be designed to express functions that address monogenic inborn errors of metabolism [119] or to exert a tumor-killing activity [120]. 

However, although FMT is a valid therapeutic approach for certain diseases, it shows some procedural limitations. The preparation of fresh or frozen fecal suspensions requires the constant and periodic presence of healthy donors which must be negative in a series of serological and microbiological screening tests [121]. Moreover, the stool samples have to be processed within six hours after collection for preserving anaerobes [121,122]. 

All these aspects render the process laborious and poorly standardized. Furthermore, it is essential to know in detail the bacterial composition of the suspension to be infused into the recipient, in order to increase FMT efficacy and especially safety. For these reasons, the therapeutic efficacy of a synthetic bacterial preparation called “Bacterial Consortium” is now under development [123,124]. This approach involves the isolation from healthy donors’ stool samples of several bacterial species normally present in the human intestinal microbiota, and thus the use of this “Bacterial Consortium”, composed by 13 microbial species, as a safe and valid alternative to donor stools [123].In conclusion, understanding the mechanisms by which the human microbiota can influence the progression of diseases, including gynecological disorders, can lead to the development of personalized approaches to shape the microbiota composition (and so its function), improving the symptoms and patients’ prognosis.

## Figures and Tables

**Figure 1 microorganisms-11-02407-f001:**
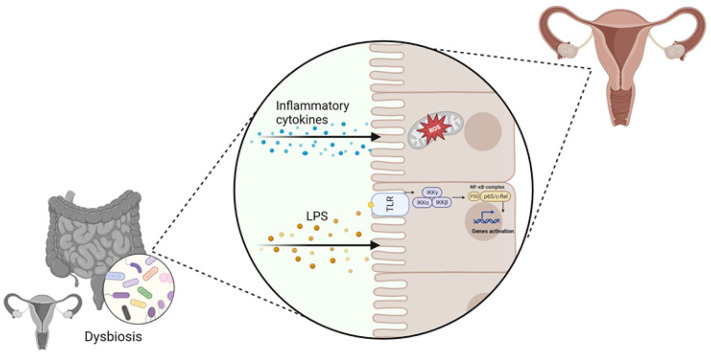
Involvement of vaginal and gut dysbiosis in the alteration of distal tissues homeostasis, contributing to gynecological disorders’ development (created with Biorender.com, accessed on 22 September 2023).

**Figure 2 microorganisms-11-02407-f002:**
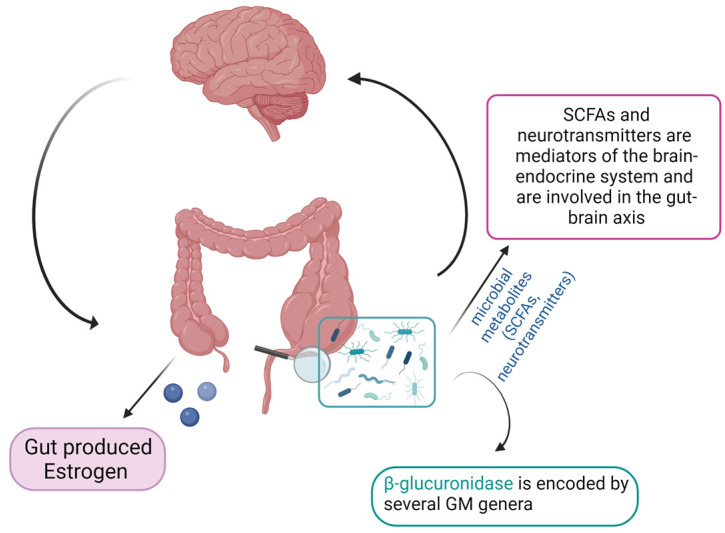
Schematic representation of the gut microbiome–estrogen axis (created with Biorender.com, accessed on 22 September 2023).

**Figure 3 microorganisms-11-02407-f003:**
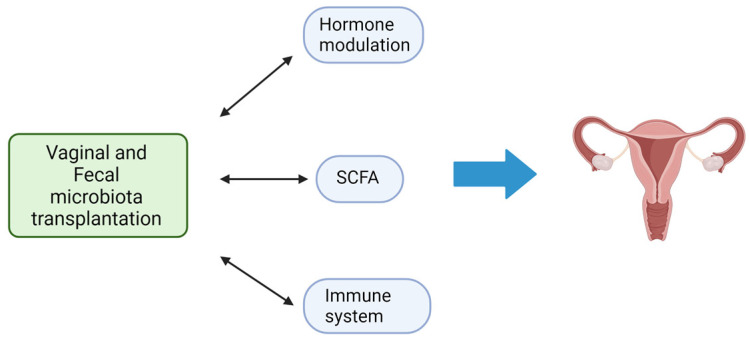
Schematic representation of the potential mechanisms of microbiota transplantation to affect gynecological homeostasis (image created with Biorender.com, accessed on 22 September 2023).

**Table 1 microorganisms-11-02407-t001:** Ongoing clinical trials involving VMT in gynecological disorders.

Study Title	NCT Number	Status	Conditions	Study Type
Vaginal Microbiota Transplant	NCT04046900	Recruiting	Recurrent Bacterial Vaginosis	Interventional
Safety and Efficacy of Vaginal Microbiota Transplant (VMT) in Women With Bacterial Vaginosis (BV)	NCT03769688	Withdraw	Bacterial Vaginosis	Interventional
Vaginal Microbiome Transplantation for Recurrent Bacterial Vaginosis	NCT04517487	Recruiting	Bacterial Vaginosis	Interventional

## Data Availability

Data sharing is not applicable to this article.

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
