# Peer review of "Microbiota Transplant and Gynecological Disorders: The Bridge between Present and Future Treatments"

_microorganisms, 2023, doi:10.3390/microorganisms11102407_

Round 1
Reviewer 1 Report
I have now reviewed the manuscript pertaining to microbiota transplant and gene disorders.
I strongly feel that the manuscript requires changes before it can be considered for accepting.
The authors need to have sections mentioning about microbiota transplant, the correlation and/or microbiome-gynecological disorders (pathways involved) before jumping to FMT.
There needs to be a figure (pathway) to make readers better understand.
I believe that the manuscript needs a section of FMT and gynecological disorders before the animal models section.
Is vaginal FMT the only major highlight for gynecological disorders? OR is the manuscript focussing entirely on VFMT? IF so, the authors need to change the title.
There are slight changes required as there are few minor typos recorded.
Author Response
POINT BY POINT
REVIEWER 1
Point 1: The authors need to have sections mentioning about microbiota transplant, the correlation and/or microbiome-gynecological disorders (pathways involved) before jumping to FMT.
Reply 1: We thank the reviewer for the right suggestion. We added the “Introduction” paragraph which includes information regarding microbiota roles and functions, comprising pathways involved in eubiosis/dysbiosis, as well as the rationale for the investigation on gynecological disorders topic. Additional connections between microbiota and gynecological disorders is reported starting from line 172.
Point 2: There needs to be a figure (pathway) to make readers better understand.
Reply 2: In agreement with the reviewer's suggestion, and for better understanding the importance of the microbiota metabolites’ functions, we have included a figure (Figure 1) describing the involvement of gut dysbiosis in the alteration of distal tissues’ homeostasis, that can favor gynecological disorders onset.
Point 3: I believe that the manuscript needs a section of FMT and gynecological disorders before the animal models section.
Reply 3: We thank the reviewer for the critical comment, but unfortunately, there are no sufficient studies on humans (as reported in line 227) to develop a specific paragraph. Nevertheless, we have included the additional “Fecal Microbiota Transplantation” paragraph to adequately report the aims and evolution of this clinical approach.
Point 4: Is vaginal FMT the only major highlight for gynecological disorders? OR is the manuscript focusing entirely on VFMT? IF so, the authors need to change the title.
Reply 4: We thank for the reviewer's comment, but we prefer to use this generic title that well mirrors the aim of the manuscript and so the subsequent discussion of both fecal and vaginal microbiota transplantation (FMT and VMT).
Point 5: There are slight changes required as there are few minor typos recorded.
Reply 5: We thank the reviewer for noticing some errors, we corrected the suggested sentences (please see see the lines: 170, 175, 264, 400). In addition, we have removed different typos’ errors such as double spaces
Finally, we have added a section including the criteria used to finalize the study, and a new figure (Figure 3) summarizing the supposed mechanisms of microbiota transplantation to favor the gynecological homeostasis.
Reviewer 2 Report
The manuscript is very interesting and novel. The authors make an up-to-date and very well-founded presentation
I. Major comments:
1. The first paragraph is good, but I think it is necessary to start the review with an introduction.
2. Include a section on methodology. It is necessary to include the criteria used to select the cited papers.
3. I suggest including the effects of the microbiota on the liver in the review. This point is important, because the benefits generated by a healthy microbiota directly influence liver metabolism. Especially in the context of obesity and NAFLD.
4. The short-chain fatty acids that are generated by the metabolism of the microbiota produce benefits in different tissues. This point should be included in the review. In addition, a better description of the different short-chain fatty acids is needed.
5. It is necessary to include dietary aspects that influence the microbiota. Especially pre and post transplant. I suggest including projections on this topic.
II. Minor comments:
1. Improve the wording of the objective or purpose of the study.
2. It would be interesting for the authors to include a figure that summarizes the expected or proposed effects, but including mechanisms that would facilitate the understanding of the manuscript.
The manuscript is well written, but some editorial errors need to be corrected.
Author Response
Dear Editor,
We thank the reviewers for their comments on the manuscript and the positive evaluation of our work. As suggested, we have modified the document that is significantly improved according to the reviewers’ suggestions.
Please find attached a revised version of the manuscript entitled “Microbiota transplant and gynecological disorders: bridge from the present and the future treatments”.
Revisions in the manuscript are in track change; we hope that the revisions in our manuscript and our accompanying responses will be sufficient to make our manuscript suitable for publication.
REVIEWER 2
Point 1: The first paragraph is good, but I think it is necessary to start the review with an introduction
Reply 1: In agreement with the right reviewer's suggestion, we have added an “Introduction” paragraph focus on microbiota roles and functions.
Point 2: Include a section on methodology. It is necessary to include the criteria used to select the cited papers.
Reply 2: We thank for the reviewer's comment. In section 2 we added the Methods section that details the criteria used to finalize the study.
Point 3: I suggest including the effects of the microbiota on the liver in the review. This point is important, because the benefits generated by a healthy microbiota directly influence liver metabolism. Especially in the context of obesity and NAFLD.
Reply 3: Thanking the reviewer for the suggestion, we have added some insights into the interplay between microbiota and several organs (such as the liver), emphasizing the role of dysbiosis in the development of various liver-related diseases.
Point 4: The short-chain fatty acids that are generated by the metabolism of the microbiota produce benefits in different tissues. This point should be included in the review. In addition, a better description of the different short-chain fatty acids is needed.
Reply 4: As rightly suggested by the reviewer, we have included a fine description of short-chain fatty acids (SCFAs) and its functions in the “Introduction” section and in the paragraph 3 (3. Role of the gut microbiome-estrogen axis in gynecological disorders) (please see the lines 153 to 155).
Point 5: It is necessary to include dietary aspects that influence the microbiota. Especially pre and post-transplant. I suggest including projections on this topic.
Reply 5: Thanking the reviewer for the critical suggestion, we have added information about nutritional roles in affecting microbiota shaping, both in the “Introduction” and “Other Microbiota-Changing Strategies and future perspectives” sections.
Point 6: Improve the wording of the objective or purpose of the study.
Reply 6: In agreement with the reviewer, we have added the aim of the study at the end of “Introduction” section (see the lines 82 to 85).
Point 7: It would be interesting for the authors to include a figure that summarizes the expected or proposed effects, but including mechanisms that would facilitate the understanding of the manuscript.
Reply 7: We thank the reviewer for the suggestion, we added two new figures: the Figure 1 describes the roles of gut dysbiosis in disrupting the balance of distant tissues, thereby contributing to the onset of gynecological disorders while the Figure 3 shows the effects by which microbiota transplantation might affect gynecological homeostasis.
Point 8: The manuscript is well written, but some editorial errors need to be corrected.
Reply 8: We thank the reviewer for noticing some errors, we corrected the suggested sentences (please see see the lines: 170, 175, 264, 400). In addition, we have removed different typos’ errors such as double spaces.